# Biogeography and Diversification of the Tropical and Subtropical Asian Genus *Gastrochilus* (Orchidaceae, Aeridinae)

**Yang Li [1], Weitao Jin [1], Liguo Zhang [1], Peng Zhou [1], Yan Luo [2], Ziwei Zhu [3] and Xiaoguo Xiang [1,\*]**

1   Jiangxi Province Key Laboratory of Watershed Ecosystem Change and Biodiversity, Institute of Life Science, School of Life Sciences, Nanchang University, Nanchang 330031, China; liyang192116@163.com (Y.L.); jin1234933@126.com (W.J.); zhangliguo1997@163.com (L.Z.); zhoupengrjcck@163.com (P.Z.)
2   Xishuangbanna Tropical Botanical Garden, Chinese Academy of Sciences, Mengla 650223, China; luoyan@xtbg.org.cn
3   Jiangxi Academy of Forest, Nanchang 330013, China; ziweiangel@163.com
\*   Correspondence: xiangxg2010@163.com

**Abstract:** Tropical and subtropical Asia are major orchid diversity and endemism centers. However, the evolutionary dynamics of orchids in these areas remain poorly studied. *Gastrochilus* D. Don, a species-rich orchid genus from tropical and subtropical Asian forests, was employed to investigate the issue. We firstly used eight DNA regions to reconstruct the phylogeny and estimate the divergence times within *Gastrochilus*. We inferred the ancestral ranges and conducted a diversification analysis based on empirical and simulated data. Subsequently, we assessed the ancestral niche state and tested for phylogenetic signals in the evolution of niche conditions. Our results suggested that the most recent common ancestor of *Gastrochilus* occurred in the subtropical area of the East Asiatic region in the late Miocene (8.13 Ma). At least eight dispersal events and four vicariant events were inferred to explain the current distribution of *Gastrochilus*, associated with the global cooling from the Plio-Pleistocene. The genus experienced a slowly decreasing diversification rate since its origin, and no significant correlation between current niches and phylogenetic relatedness was observed. The diversification of *Gastrochilus* was attributed to accumulation through time, integrated with the intensification of the Asian Monsoon system during the Plio-Pleistocene, pollination, and epiphytism.

**Keywords:** *Gastrochilus*; biogeography; diversification; niche; Asian Monsoon





## 1. Introduction

Exploring the patterns of plant diversity today is a basic issue for biogeographers and evolutionists. To better understand the dynamics of plant diversity, it is vital to integrate the historical biogeography and the niche requirements of species [1]. Over the past two decades, the phylogenetic niche conservatism (PNC) and niche evolution (NE) hypotheses were proposed to account for species diversity [2,3]. PNC deems that most species tend to maintain their ancestral niches, survive in similar climatic environments, and differentiate in-situ [4]. Several studies have shown that the PNC hypothesis may explain the diversity of groups with different evolutionary histories [5–8]. For example, the diversity patterns of Zygophyllaceae at the global scale can be attributed to the strong phylogenetic conservatism in their precipitation-related niches [7]. In turn, NE posits that species may expand their niche breadth or occupy new conditions and can diversify in new habitats and climatic regimes [2]. For example, the diversity of *Hakea* (Proteaceae) in different geographic regions was explained by the frequency of evolutionary biome shifts [9]. Although there are many studies strongly supporting the role of PNC or NE in explaining clade diversification [4,7,9–11], several researchers suggested that there was no significant correlation between current niches and phylogenetic relatedness [12,13], especially the organisms in isolated habitats such as the birds in high-altitude regions [14], which implied that the species diversity was attributed to accumulation through time.

Tropical and subtropical Asia are major orchid diversity centers [15,16]. These regions are characterized by their high plant diversity and endemism [17,18] and have also been considered both a "Cradle" and a "Museum" for vascular plants since the Cretaceous [17,19,20]. During the Cenozoic, the Asian mainland experienced a series of complex geological and climate changes, such as the uplift of the Himalaya-Tibetan Plateau [21] and the establishment and intensification of the Asian monsoon [22]. The multistage uplifts of the Himalayas resulted in significant climate changes, new geophysical environments, novel ecological niches, and the formation of physical and physiological isolation barriers across the faunal and floral elements of Asia [23,24]. On the one hand, the uplift of the Himalayas provided many new niches and is attributed to organisms' diversification [25]. On the other hand, the four periods of intensification of the East Asia Summer Monsoon (EASM) during the Cenozoic have possibly brought abundant rainfall [26] and are positively correlated with plant richness [27–30]. Therefore, the uplift of the Himalayas and the EASM produced more new niches and conditions for organisms and are proposed to explain biological diversification in East Asia [29–31].

*Gastrochilus* D. Don (1825) (Aeridinae, Vandeae, Epidendroideae, Orchidaceae) is an epiphytic orchid genus widely distributed in tropical and subtropical Asia [15,32]. Thanks to its high morphological diversity and brightly colored flowers, it has potential horticultural value [15]. Since the latest preliminary revision of *Gastrochilus* [32], nearly 20 new species have been found in south China (Chongqing, Yunnan, Taiwan), Vietnam, Myanmar, Nepal and India [33–45]. Additionally, *Haraella* Kudo and *Luisiopsis* C.S.Kumar and P.C.S.Kumar have been transferred to *Gastrochilus* [46,47]. Therefore, the genus *Gastrochilus* consists now of 69 species, of which many are narrow endemics, with a species diversity center in the South-East Asian archipelago [36,48,49]. Recently, Liu et al. [49] revealed that *Gastrochilus* is monophyletic and divided into five clades based on five DNA regions (ITS, *matK*, *psbA-trnH*, *psbM-trnD*, *trnL-F*) and inferred that pollination system shifts in *Gastrochilus* have occurred independently at least three times. Liu et al. [50] reconstructed the phylogenetic relationships within the *Cleisostoma–Gastrochilus* clades (Aeridinae) based on the complete chloroplast genome, strongly supporting the monophyly of *Gastrochilus*. However, the spatio-temporal evolution of the genus is still unclear.

In this study, our objectives are (1) to estimate divergence times within *Gastrochilus* using eight plastid and nuclear DNA regions, (2) to investigate the historical biogeography of *Gastrochilus*, and (3) to explore the factors that have led to its diversification.

## 2. Materials and Methods

### 2.1. Taxon Sampling and Molecular Data

In this study, we sampled 34 species of *Gastrochilus*, comprehensively covering the distribution range of this genus. Based on Pridgeon et al. [15] and Farminhão et al. [51], 18 species closely related to *Gastrochilus* from Aeridinae and four species from Angraecinae were used as outgroups. All sequence data were downloaded from the GenBank (https://www.ncbi.nlm.nih.gov/ (accessed on 19 December 2021)). The phylogenetic analysis of Epidendroideae showed that a proportion of potentially parsimony informative sites of the internal transcribed spacer (ITS) *trnL-F*, *matK* and *rbcL* were 64%, 28%, 28%, and 11%, respectively, and they showed their strong ability to resolve species relationships [52]. It has been suggested that the addition of non-coding chloroplast regions could provide higher relative variability in resolving species relationships [53,54]. A total of eight DNA markers were employed in this study, including one nuclear marker (ITS) and seven chloroplast DNA markers (*atpH-I*, *matK*, *psbA-trnH*, *psbM-trnD*, *rbcL*, *trnL-F*, and *rps19-rpl22*). Taxon information and GenBank accession numbers are listed in Table S1.

### 2.2. Phylogenetic Analysis

DNA sequences were aligned and subsequently manually adjusted in BioEdit [55]. Topological congruence between the chloroplast and nuclear data was evaluated using the incongruence length difference (ILD) test [56]. The partition homogeneity test for plastid

DNA and ITS shows character incongruence ($p$ = 0.01). Visual inspection indicates that there are very few "hard" conflicts between the plastid vs. ITS trees, and such conditions have been interpreted by Wendel and Doyle [57] as a soft incongruence, which might disappear with additional data. Gatesy et al. [58] demonstrated that concatenating truly incongruent data sets could still increase resolution and branch support. Therefore, we combined the datasets for subsequent analyses. All characters were unordered and had equal weight. Gaps were treated as missing data.

Three phylogenetic reconstruction methods were performed, including maximum parsimony (MP), maximum likelihood (ML), and Bayesian inference (BI). MP analyses were performed in PAUP* 4.0b10 [59]. Heuristic searches were conducted with 1000 replicates of random addition, in combination with tree-bisection-reconnection (TBR) branch-swapping, MulTrees in effect, and steepest descent off. Bootstrap support values were conducted with 1000 replicates with 10 random taxon additions and heuristic search options.

Based on the Akaike information criterion (AIC), the best-fit nucleotide substitution model of DNA regions was chosen using ModelTest v.3.7 [60]. ML analyses were conducted in RAxML v.8.4 [61]. We conducted a rapid bootstrap analysis (1000 replicates) and searched for the best-scoring ML tree simultaneously. BI analyses were performed in MrBayes v.3.2 [62]. Four Markov Chain Monte Carlo tests were run, sampling one tree every 1000 generations for 3,000,000 generations. Tracer v.1.5 was used to assess chain convergence and ensure that the effective sample sizes (ESS) are above 200 for all parameters [63]. Majority rule (>50%) consensus trees were constructed after removing the "burn-in" samples (the first 20% of the sampled trees).

### 2.3. Time Estimation

Firstly, the likelihood ratio test (LRT) [64] was conducted to determine whether the data evolved in a clock-like fashion. Log-likelihood ratios of the clock and non-clock model were compared. The degree of freedom is equivalent to the number of terminal taxa minus two, and significance was assessed by comparing two times the log-likelihood difference to a chi-square distribution [65]. The LRT test rejected a clock-like evolution ($\delta$ = 1363.8014, df = 54, $p < 0.001$), and therefore. we used a relaxed lognormal clock model to estimate the divergence in BEAST v.2.6.0 [66]. There is no *Gastrochilus* fossil nor any fossils of one of its close relatives in Aeridinae and Angraecinae; thus, two calibration points were set based on Givnish et al. [16,67]: (1) the split age of Aeridinae and Angraecinae (21.21 Ma) was used for the tree root age, and a prior normal distribution (SD = 3.05) for the calibration point was assigned following the suggestion of Ho [68]; (2) the crown age of Aeridinae was set to 16 Ma with a normal distribution (SD = 1.0). The speciation prior was set as YULE, and the substitution model of DNA regions was selected as the GTR+I+Γ model. Markov Chain Monte Carlo (MCMC) searches were run for 100,000,000 generations and sampled every 1000 generations. Convergence was assessed by Tracer v.1.5 [63], and the effective sampling size for all parameters was >200. The maximum clade credibility (MCC) tree was computed by TreeAnnotator v.1.7.4 [69].

### 2.4. Biogeographical Analyses

Based on the extant distribution of *Gastrochilus* and outgroups, four main regions were categorized based on Takhajan [70]: East Asiatic region (A), Indian region (B), Indo-Chinese region (C) and Malesian region (D). The ancestral range reconstruction was inferred using the Statistical Dispersal–Extinction–Cladogenesis (S-DEC) model, as implemented in RASP [71]. In S-DEC, it summarizes biogeographic reconstructions across all user-supplied trees. The DEC model is applied to each ultrametric tree within a posterior distribution resulting from a Bayesian phylogenetic analysis. Subsequently, we calculated the probability of an ancestral range x at node n on a summary tree [71]. The MCC tree obtained from BEAST was chosen as the summary tree. The random 1000 trees from BEAST trees after burn-in were input to estimate probabilities of ancestral range at each node.

### 2.5. Ancestral State Reconstruction and Correlates of Diversification

Ancestral state reconstruction was performed using the maximum likelihood method implemented in BayesTrait v.4.0 [72]. Information about species habitats was compiled from online databases (www.gbif.org (accessed on 23 January 2022); www.orchidspecies.com (accessed on 23 January 2022); www.africanorchids.dk (accessed on 6 March 2022); www.iplant.cn (accessed on 10 January 2022)), and the taxonomic literatures [32–45], and we defined two states: (1) tropical (state 0); (2) subtropical (state 1) (Table S2).

The binary state speciation and extinction model (BiSSE) was used to examine whether the climatic zone is directly correlated with differential rates of diversification implemented in DIVERSITREE 0.9-6 [73]. To correct for non-random, incomplete sampling, we specified sampling fractions, i.e., the proportion of species in tropical Asia and in subtropical Asia that are included in the tree.

### 2.6. Diversification Analysis

Birth–death likelihood (BDL) models were used to test the significance of heterogeneity or the consistency of the temporal diversification rate [74]. The model selection was based on the difference in the AIC scores between the best-fitting rate-constant and rate-variable models ($\Delta AIC_{RC}$). The calculations were performed using laser 2.3 [74].

To better understand diversification rates in *Gastrochilus*, we employed two methods to analyze rates. First, semi-logarithmic lineage-through-time (LTT) plots were constructed using the R package ape 2.5-1 [75]. The MCC tree was used to generate the tempo of diversification, and 1000 trees were sampled randomly from the converged BEAST trees to calculate a 95% credibility interval. Second, we used CLaDS (cladogenetic diversification rate shift model), a model-based approach to estimate speciation rates [76]. CLaDS applies a Bayesian approach to infer speciation rates along a phylogeny and assumes that rates change after every speciation event.

To evaluate the effect of the missing species, we add all 35 missing species randomly in the MCC tree in the R package 'phytools' 0.4-60 [77]. Then we carried out a diversification analysis in LTT and CLaDS.

### 2.7. Collection of Species Distribution Data and Environmental Variables

Distribution data of *Gastrochilus* were collected from online databases (the global biodiversity information facility, https://www.gbif.org/ (accessed on 23 January 2022)), herbaria (Herbarium, Institute of Botany, Academia Sinica (PE), and Herbarium of Jiangxi University (JXU)), and our fieldwork. These datasets were carefully assessed, and some erroneous records (i.e., occurrences in the oceans, ice sheets, and deserts), duplicates, and cultivation records were removed. Finally, a total of 262 unique distribution records from 33 species were used in this study (Table S3). We also collected 20 environmental variables including 19 bioclimatic variables and one topographical layer (elevation) (https://www.worldclim.org/data/worldclim21.html (accessed on 22 January 2022)) [78]. All environmental variables are at a resolution of 30 arc seconds. Mean values of the variables for each species were used in the further analysis.

### 2.8. Estimation of Evolutionary Rate in Niche Traits

To estimate the evolutionary rate of niche in *Gastrochilus*, we firstly ordinated all environmental variables and niche data using phylogenetic principal component analysis (PCA) implemented in R package phytools 0.7-70 [77] with the "phyl.pca" function. Then, we conducted complementary runs using the BAMM trait model on the first axis of the phylogenetic PCA of niche traits in BAMM 2.5.0 [79]. For niche rate, the MCMC was run for 10 million generations and sampled every 5000 generations. Prior values were selected using the "setBAMMpriors" function. Postrun analysis and visualization used the R package BAMMtools 2.17 [79]. The initial 25% of samples of the MCMC run were discarded as burn-in, and the remaining data were assessed for convergence using the CODA package [80] to ensure that the ESS values were above 200.

*2.9. Detection of Phylogenetic Signals of Niche Traits*

Phylogenetic signals were measured using Blomberg's $K$ [81] and Pagel's $\lambda$ [82]. We estimated the phylogenetic signals based on the time-calibrated tree and the first axis of the phylogenetic PCA of niche traits of *Gastrochilus* using the "phylosig" function in the R package phytools 0.7-70 [77].

## 3. Results

*3.1. Phylogenetic Relationships and Divergence Time Estimates within Gastrochilus*

The total length of combined DNA sequences was 10,065 bp, of which 1351 characters were variable, and 551 characters were parsimony informative. The monophyly of *Gastrochilus* was strongly supported (BI-PP = 1.00, ML-BP = 97, MP-BS = 95; Figure 1). The inter-species relationships within *Gastrochilus* were supported by moderate to high supporting values, but the relationships among the main clades have not been resolved (Figure 1).

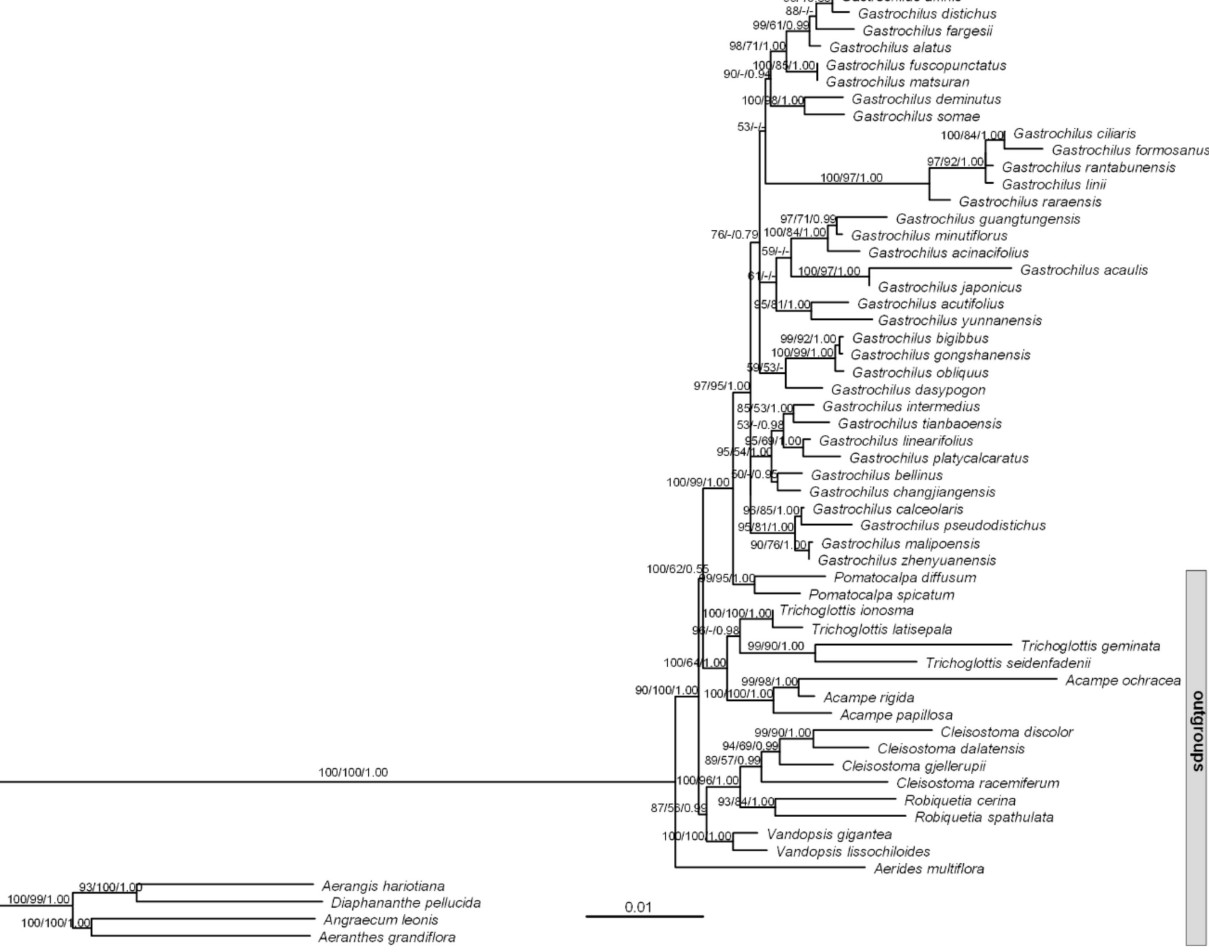

**Figure 1.** Phylogenetic tree obtained by the maximum likelihood method of the combination of nuclear and plastid regions. Numbers above the branches indicate supported values (>50%) from maximum likelihood, maximum parsimony, and Bayesian Inference methods, respectively. Numbers at the nodes are bootstrap percentages and Bayesian posterior probabilities, respectively. A dash (-) indicates that a node is not supported in the analysis.

Our results suggest a stem age of *Gastrochilus* at 9.49 Ma (95% highest probability density (HPD): 6.61–12.55; Figure 2a, node 1), and a crown age of *Gastrochilus* at 8.13 Ma (95% HPD: 5.51–10.83; Figure 2, node 2). Most of the species originated during the Pliocene and early Pleistocene (Figure 2a).

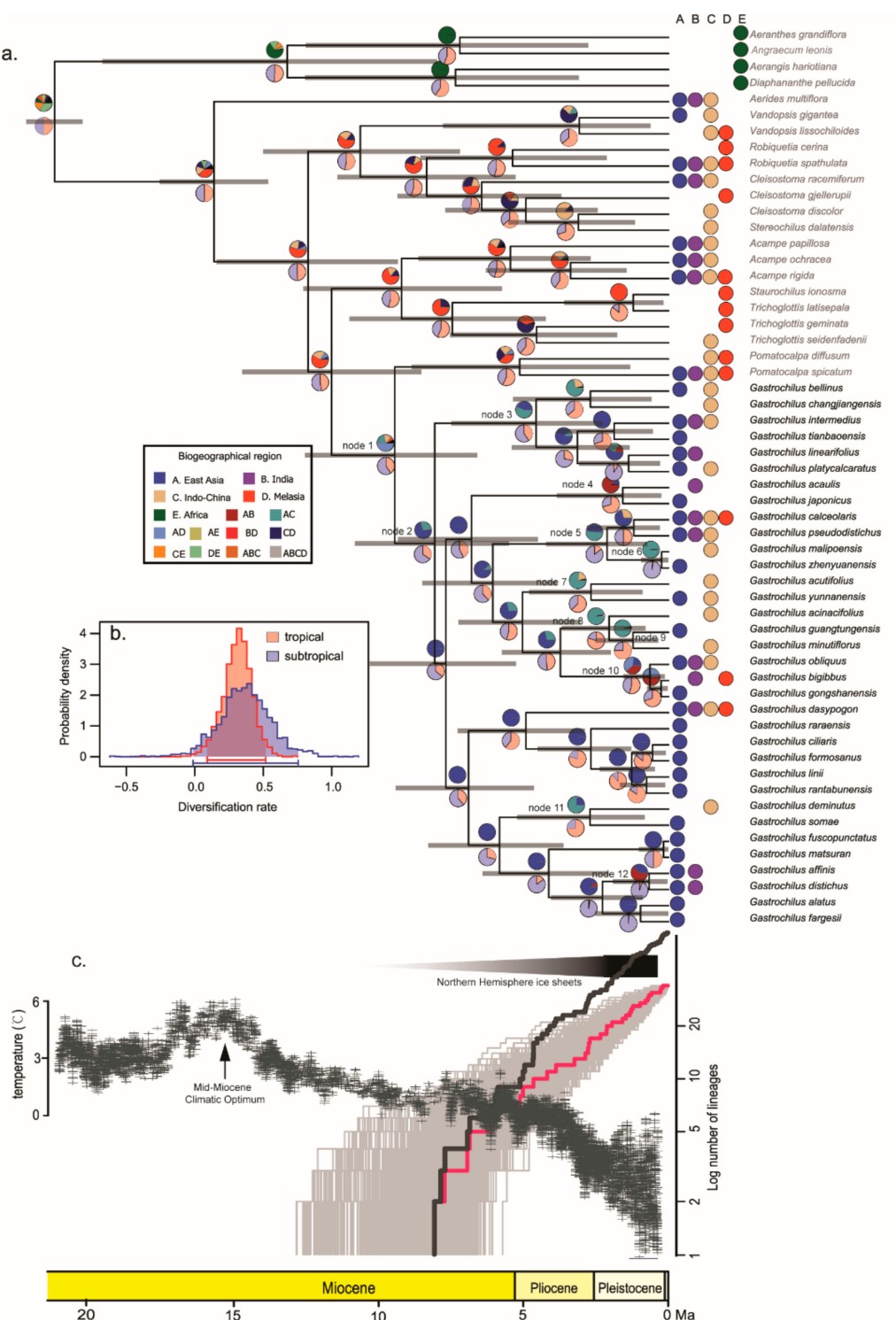

**Figure 2.** (**a**) Ancestral range reconstruction of *Gastrochilus* based on the chronogram. The chronogram was generated in BEAST analysis. Grey bars show 95% highest posterior density intervals. Nodes of interest were numbered from 1 to 12. The pie charts above the branches represent the results of ancestral range reconstruction, and those under the branches represent the results of habitat reconstruction. (**b**) Habitat-dependent posterior probability distribution of net diversification rates from BiSSE analyses. (**c**) LTT plots of *Gastrochilus* for empirical data (pink line) and simulated data (black line), respectively. The depiction of temperature changes is modified from Zachos et al. [83].

### 3.2. Ancestral Range Reconstruction

The ancestral area reconstruction of *Gastrochilus* based on S-DEC is shown in Figure 2a. The ancestral area of *Gastrochilus* is uncertain, although it probably originated in East Asia or Malesia (node 1). The range of the most recent common ancestor (MRCA) of the genus is inferred in East Asia (node 2). The current distribution of *Gastrochilus* is inferred to be the result of eight dispersal events and four vicariant events. There are three dispersal events from the East Asiatic region to the Indian region at 1.79 Ma (95% HPD: 0.25–3.86; node 4), 0.74 Ma (95% HPD: 0.13–1.56; node 10), and 0.80 Ma (95% HPD: 0.03–1.92; node 12), respectively. The remaining five dispersal events from the East Asiatic region to Indo-Chinese region happened at 4.70 Ma (95% HPD: 2.27–7.50; node 3), 2.29 Ma (95% HPD: 0.73–4.23; node 5), and 2.79 Ma (95% HPD: 0.89–4.81; node 7), 2.15 Ma (95% HPD: 0.79–3.68; node 8), and 2.86 Ma (95% HPD: 0.82–5.23; node 11), respectively. Additionally, there are three vicariant events that happened between East Asiatic and Indo-Chinese regions at 0.33 Ma (95% HPD: 0–0.95; node 6), 1.33 Ma (95% HPD: 0.35–2.52; node 9), and 2.86 Ma (95% HPD: 0.82–5.23; node 11), respectively. Only one diverged event occurred between East Asiatic and Indian regions (1.79 Ma, 95% HPD: 0.25–3.86; node 4).

### 3.3. Diversification of Gastrochilus

A positive $\Delta AIC_{RC}$ value suggests that the data are best approximated by a rate-variable model of diversification [74], so the BDL analysis rejected the null hypothesis of temporally homogeneous diversification rates within *Gastrochilus* ($\Delta AIC_{RC}$ = 2.07). The BiSSE analysis indicated that the tropical Asian lineages and the subtropical Asian lineages presented the nearly same diversification rate (Figure 2b). The LTT plot showed that *Gastrochilus* exhibited a high rate of lineage accumulation since its divergence and then decreased slowly through time (Figure 2c, red line). Furthermore, the ClaDS showed there was no significant mean speciation rate shift during its evolutionary history, and the mean speciation rate decelerated very slowly from 8.13 Ma (95% HPD: 5.51–10.83) to the present (Figure 3c,d (green line)). The simulated analysis of LTT plots and CLaDS analysis also showed that the diversification rate and speciation rate increased at the early evolutionary stage and then decreased since the latest Miocene (Figure 2c (black line), Figure 3d (black line)), respectively.

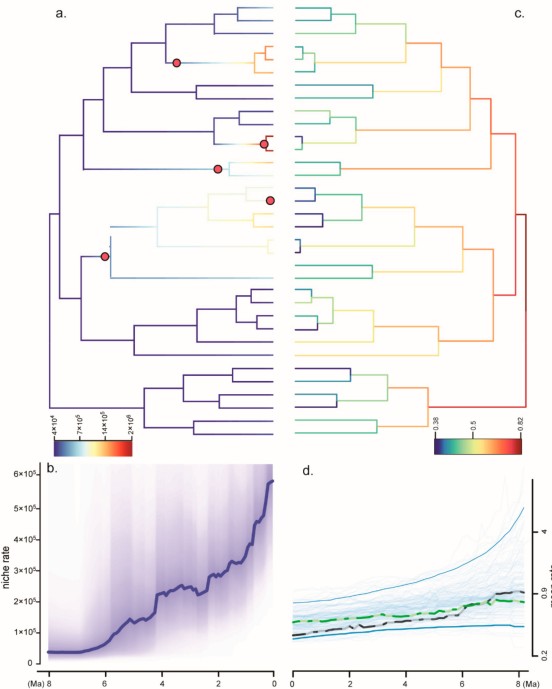

**Figure 3.** Niche analysis and diversification analysis of *Gastrochilus*. (**a**) Niche evolution and shift of

*Gastrochilus*. (**b**) Niche rate during the evolutionary history of *Gastrochilus*. (**c**) Inferred lineage-specific speciation rates for *Gastrochilus* phylogeny. (**d**) Inferred mean speciation rate of *Gastrochilus* through time, with individual MCMC iterations (thin blue line), the 95% credibility interval for each time point (thick blue line), the mean rate for each time point (dotted green) of empirical data, and the mean rate for each time point (dotted black) of simulated data. The unit of the diversification rate is speciation events per million years; niche rates are unitless.

### 3.4. Niche Evolution and Phylogenetic Signals

PC1 had a higher contribution from Bio12 (Annual precipitation), Bio16 (Precipitation of wettest quarter), elevation, Bio2 (Mean diurnal range), Bio18 (Precipitation of warmest quarter) and Bio13 (Precipitation of wettest month) (Table S4). Annual precipitation is the main influencing factor with a high PC1 loading value of about 0.88 (Table S4). Nine major shifts of the evolutionary rate of environmental factors were found in the genus *Gastrochilus*, six of which occurred in the last 2 Ma (Figure 3a). However, evolutionary rates of niche experienced a strong increase toward the present, beginning around the early Pliocene (Figure 3b). Notably, the annual precipitation seemed to play a key role in the rate shift of niche evolution (Table S4). The values of Blomberg's $K$ and Pagel's $\lambda$ are 0.175 and 0.023 with $p > 0.05$, respectively. This suggests that no significant phylogenetic signals were detected in the niche traits of *Gastrochilus*.

## 4. Discussion

### 4.1. Temporal and Spatial Mode of Gastrochilus

In the present study, our result strongly supported that *Gastrochilus* is monophyletic, which is consistent with the previous studies [49,50]. The S-DEC result inferred that the MRCA of *Gastrochilus* lived in the East Asiatic region (Figure 2a, node 2). The ancestor of *Gastrochilus* has migrated from the East Asiatic region to its adjacent regions (Indian region and Indo-Chinese region) since the early Pliocene at least eight times (Figure 2a). Following the middle Miocene climatic optimum at approximately 15 Mya, a period of global cooling began from ~11 Mya followed by the drastic temperature fluctuations during the Pliocene and Pleistocene [83]. The cooling climate caused many species to migrate southward or to lower altitudes [84–86], which also meant *Gastrochilus* dispersed from the East Asian region into the Indian and Indo-Chinese regions. However, it is well-known that the monsoon system (South Asian Summer Monsoon (SASM) and East Asian Summer Monsoon (EASM)) strengthened in the Asian mainland during the Miocene and Pliocene, especially between c. 15–4 Ma [87–89]. Recent studies reported that both South and East Asian Summer monsoons played a decisive role in the landscape evolution of the Himalayas and the adjoining areas in the Indo-Malayan Realm [90]. The intensifications of the EASM during the Late Cenozoic brought abundant rainfall and, therefore, significantly promoted the survival and differentiation of plants in tropical and subtropical Asian mainland [19,28–30,91]. During the dynamic evolutionary processes of *Gastrochilus*, five of eight migration events occurred from the East Asiatic region to Indo-Chinese region in the Pliocene to the early Pleistocene (4.70–2.15 Ma; Figure 2a, node 3, 5, 7, 8, 11), in agreement with the timing of the intensification of the EASM (3.6–2.6 Ma) [26] and SASM (3.57–2.78 Ma) [92]. Additionally, the other three events from the East Asiatic region to the Indian region happened during the late Pleistocene (1.79–0.74 Ma; Figure 2a, node 4, 10, 12), which is consistent with major changes in the monsoon cyclicity that occurred through the Mid-Pleistocene Transition between c. 1.25 and 0.7 Ma [89]. In addition, due to the drastic decline of temperature since the late Pliocene and the frequent temperature fluctuations during the Quaternary, the lineages of *Gastrochilus* experienced four vicariant events immediately (Figure 2a, node 4, 6, 9) or after dispersal events (Figure 2a, node 11). In general, the current distribution pattern of *Gastrochilus* does not appear to have occurred via long-distance dispersal. Rather, range expansions associated with a few vicariances are suggested here to explain this pattern.

*4.2. Diversification and Niche Evolution of Gastrochilus*

Our LTT plots indicate that the lineages of *Gastrochilus* had accumulated over its evolutionary time since it diverged from the sister groups (8.13 Ma, 95% HPD: 5.51–10.83; Figure 2c). The rate-through-time plots by CLaDS suggested that the mean speciation rates of *Gastrochilus* decreased slowly through its evolutionary history (Figure 3c,d). The diversification analyses of simulated data also indicated the same tendency of its evolutionary dynamics (Figures 2c and 3d). The same diversification pattern has been detected in *Cirrhopetalum* alliance (*Bulbophyllum*, Orchidaceae) [93]. Since the late Pliocene, the global cooling has intensified [83,88], and it might have brought about the slow speciation rates of *Gastrochilus*.

Although there are nine significant niche shifts in the evolutionary history of *Gastrochilus* (Figure 3a), both the values of Blomberg's $K$ (0.75, $p > 0.05$) and Pagel's $\lambda$ (0.023, $p > 0.05$) inferred that there are no significant phylogenetic signals in the niche traits of *Gastrochilus*. Furthermore, both the lineages in tropical Asia and subtropical Asia demonstrated a similar diversification rate (Figure 2b), although more than 65% of currently recognized species are restricted to tropical Asia. Our results imply that the species diversity of *Gastrochilus* is explained by accumulation through time. This result is similar to the diversification pattern of *Bulbophyllum* in tropical and subtropical Asia, in which species richness is most likely the result of a time-for-speciation effect since the late Miocene [94]. Our environmental niche analyses demonstrated that annual precipitation is an important environmental variable determining the distribution of *Gastrochilus* (Table S4). Statistically, more than 70% of the extant diversity within this genus was generated in the late Pliocene and the Early Pleistocene (Figure 2a). The intensifications of the EASM during the Pliocene brought abundant rainfall to the tropical and subtropical Asian mainland and probably facilitated the diversification of *Gastrochilus* with numerous dust-like seeds. Moreover, Givnish et al. [16] proposed that the remarkable diversity of orchids is apparently driven in part by the acquisition of pollinia, epiphytism, tropical distributions, CAM photosynthesis, pollination syndromes, and life on extensive tropical cordilleras. They also pointed out that shifts in net diversification are scale-dependent, and multiple factors—several of them interconnected—have contributed to orchid diversification at the genus level. Liu et al. [49] showed that the presence of epichile hairs has switched many times in *Gastrochilus*, representing a character state evolving as an adaptation to bee pollination [95], and thus speculated that pollination system shifts occurred independently at least three times in *Gastrochilus* [49]. The pollination shifts in promoting speciation are recorded in *Holcoglossum* [96]. Furthermore, except for a few species growing on rocks (e.g., *G. gongshanensis*), the genus is mainly found on the tree trunks in rainforests, broadleaved forests, or coniferous forests (Table S2) [37,48,50]. In a word, the diversification of *Gastrochilus* is not only a result of the intensification of monsoons in the last c. 10 Ma but is also attributed to the integration of pollination syndromes and epiphytism.

**Supplementary Materials:** The following are available online at https://www.mdpi.com/article/10.3390/d14050396/s1, Table S1: The samples and GenBank accession numbers used in this study, Table S2: The distribution, climatic zone, phenology, and habitat of species in this study, Table S3: The species distribution data used in this study, Table S4: Niche PCA loadings.

**Author Contributions:** Y.L. (Yang Li), Z.Z. and X.X. designed the research. P.Z., Y.L. (Yan Luo), L.Z., W.J. and X.X. collected and performed analyses. W.J., X.X. and Z.Z. drafted the manuscript, W.J., X.X., Y.L. (Yang Li), Y.L. (Yan Luo), P.Z., L.Z. and Z.Z. revised the manuscript. All authors have read and agreed to the published version of the manuscript.

**Funding:** This research was funded by the National Natural Science Foundation of China–Yunnan Joint Fund Project (grant number: U1802242), the National Natural Science Foundation of China (grant numbers: 31670212, 31300181, and 32060056), and Guangxi Key Laboratory Construction Project (grant number: 19-185-7).

**Institutional Review Board Statement:** Not applicable.

**Data Availability Statement:** Not applicable.

**Conflicts of Interest:** The authors declare no conflict of interest.

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
