# Peer review of "Biogeography and Diversification of the Tropical and Subtropical Asian Genus Gastrochilus (Orchidaceae, Aeridinae)"

_diversity, doi:10.3390/d14050396_

Round 1

Reviewer 1 Report

refer to attachment file

Author Response

    We have made a point-by-point response(changes) to the  reviewers' comments marked on revision.

Reviewer 2 Report

The manuscript "Biogeography and diversification of the tropical and subtropical Asian orchid Gastrochilus D. Don" is interesting and could be published, but I recommend a set of topics that authors should mention to support the knowledge about this genus and the interpretation od the results. It is possible that some of these topics are unknown, but I think that authors must indicate some paragraphs about them.

First of all, the data about the ecology are too few or absent, regarding the altitude, the type of vegetation and the characteristics of the trees in which the species grow. In the discussion, a general mention is found: "epiphytic on the tree trucks [trunks] of tropical and subtropical forests in Asia, with a few species on the rocks". I recommend to add a Table with these data.

Secondly, no information about pollination is provided, probably for the absence of data, but a paragraph about this topic in the Discussion would be included.

Thirdly, data about phenology and seed dispersal would be provided. Is there any relationship between monsoons and reproductive biology?

Finally, the distribution of the species in Table S1 is not complete. Some species are currently present in more countries that those indicated in the Table. I recommend to include more distribution maps concerning the clades in Figure 3.

In the attached file, I include some corrections of the text.

Author Response

We have made a point-by-point response to the your comments, and please see the attached file.

Reviewer 3 Report

In this manuscript the authors made use of published DNA sequence data obtained from a public database (GenBank) and performed several phylogenetic methods that are usually employed to test hypotheses about the diversification of clades.  I believe the phylogenetic methods were properly performed for the most part (although, see comment below on molecular clocks and use of MCC tree).  Data were collected on species distribution from GBIF database and the environmental variable were also obtained from a public repository.  This seems a nice way to make use of and analyze available data.  I raise a few points related to the work below. 

1-The main comment is that it seems like the authors have accessed all of these databases and then just used every phylogenetic method available to see what results could be obtained.  There were no hypotheses presented or justification for why the hypotheses should be considered in the first place.  Perhaps because the authors just made use of public data, they do not know the biology of Gastrochilus very well.   For example, on lines 145-147 it is stated “Ancestral habitat reconstruction was performed using the ML method implemented in Mesquite 2.7.4 under the Mk model [68]. Information about species habitats was compiled from their distributions, and we defined two states: (1) tropical Asia (state 0); (2) subtropical Asia (state 1).   First of all, how are the distinctions made between these two areas, and how are those possibly considered “habitats’?   All Gastrochilus are epiphytes so what are the authors really investigating here?  There is no justification for investigating this question.  From Fig. 3 it looks like there are probably only 2/3 species that may be “subtropical”.  There are no data shown for the outgroups in terms of whether they are tropical or subtropical but from my experience most are tropical. 

2-The authors then use BISSE to ask if tropical or subtropical distribution is related to diversification rate.  Here again, there is no justification for why this question is investigated for the genus Gastrochilus.  Why should this be hypothesized?  There should be a very good reason especially since BISSE is well-known to be unreliable without HUGE and complete datasets.  Although the authors have attempted to correct for the large number of missing species, I don’t believe meaningful results will be obtained with this method on this small dataset.  IF only 2-3 species are subtropical, why would we want to investigate this question????

3-Next the authors look at more diversification of Gastrochilus using additional methods (LTT plots and CLaDS).   Again, it is not clear why this question is being asked.  Why should be hypothesize that Gastrochilus has diversified differently throughout time?

4-Figure 3 is VERY difficult to interpret.  First of all, no key is provided to help a reader interpret the ranges A-ABCD.  We are forced to go back to the methods section and read what they represent.   Then for the LTT plot, it is not clear what the different color lines represent.  The legend does not explain what the pink and black line are showing.  Presumably the grey lines are the simulations but why do the pink and dark black line differ and what are they

5-On lines 224-226 the authors state:  “The current distribution of Gastrochilus is inferred to be the result of 8 dispersal events.  There are three dispersal events from the East Asiatic region to Indian region at 1.76 Ma  (95% HPD: 0.29−4.2; Figure 3, node 4), 0.74 Ma (95% HPD: 0.16−1.76; Figure 3, node 8), and 0.74 Ma (95% HPD: 0.02−2.16; Figure 3, node 11), respectively. “  It is unclear to me how the authors can conclude this.  First of all, why aren’t these distributions the result of vicariance events instead of dispersals?  Secondly, at node 8, the species are found in multiple areas.  How can we conclude that there was a dispersal to India?  The same question stands for node 11.    

6-On lines 240-244  it is stated “Furthermore, net diversification rate estimated by ClaDS showed no shift and decelerated very slowly from 8.94 Ma (95% HPD: 5.85−12.06) to present in Gastrochilus (Figure 4c, d). The simulated analysis of LTT plots and CLaDS also showed that there was diversification rate increased at the  early evolutionary stage, and then decreased drastically since the latest Miocene (Figure  3c, Figure 4d).”   The conclusions seem at odds here:  how can the authors conclude that there was no shift and a slow deceleration in the first two lines but then conclude a “drastic” decrease in diversification rate in the last line?

7-Given these comments, the Discussion section about the biogeography of Gastrochilus needs to be revised because of apparent contradictions.  For instance on lines 289-293 the authors claim there has NOT been dispersal but in the results section that is all that is described.  The authors then attempt to tie their results to patterns of climatic change during the Pliocene.  Yet, without knowing the specific distributions of each species, it is very difficult to interpret this as a reader.  I suggest that the authors provide range maps for each species in a supplemental document.  That way, we can interpret the results.   As it stands now, Fig. 1 is not useful.

8-On lines 299-301 it is stated: “The diversification analyses of simulated data also indicated the same tendency of its evolutionary dynamics (Figure 3c, Figure 4d). Since the late Pliocene, the global cooling intensified [78], there may not only promote the Gastrochilus migrate southward, but also slowed speciation rates [81].”  This makes absolutely no sense based on the results presented.  The authors claim that the genus originated in tropical asia (lines 235-236) and then migrated to subtropical areas.  So, why then should they claim there was a southward movement from subtropical areas to tropical ones??   

Overall, the Discussion is lacking any clear ties between the results of the analyses performed and the climatic changes over the last 4 million years. 

9-124-125:  The authors state “Firstly, a likelihood ratio test was conducted to determine whether the data were  evolved in a clock-like fashion. The result rejected a constant rate (-ln L = 23318.64123, d.f. = 51, P < 0.001).” It is not clear how a strict clock can be rejecxted since the authors only report the -lnL of the tree.  To do this requires use of a test statistic (presumably chi-square?)and that is not provided.  What was the likelihood of the non-clock tree hypothesis? 

10-Lines 141-142:  The authors state: “The MCC tree obtained from BEAST was chosen as the input tree. The random 1000 trees from BEAST trees after burn-in were input to estimate probabilities of ancestral range at each node. “  However, it is not clear from the manuscript whether one a single tree was used (which is what the first sentence implies) or a set of trees from the posterior distribution (which might be what the second sentence implies).  If indeed a set of trees were used, which analyses used them and how were their results summarized?

Author Response

(The authors gave the same response as above.)

Reviewer 4 Report

It is a very good and solid manuscript on a tropical orchid genus. The analysis is diverse and the choice of analysis methods is justified. But after reading through the manuscript I have a feeling of superficiality. In the discussion I miss the details about the species of the genus. The distribution map (Figure 1) is also too general. Where are the Study Accessions Spread on the Map? I think the most important information about examined accessions hiding in Supplement (Table S1) is wrong.
It is also necessary to mention that the most important goals are carried out in the manuscript.
I only find one misprint in the text. Page 7 line 235: tropical Aisa - must be Asia.

Author Response

(The authors gave the same response as above.)

Reviewer 5 Report

The authors apply a series of standard phylogenetic comparative methods to investigate the evolutionary dynamics of Gastrochilus.

The overall approach is interesting, but the most significant results rely on a dated phylogeny, whose calibration is debatable. The authors should explain better how the calibration point of 16 Mya was chosen and why they did not include at least an angraecoid orchid in the outgroup to improve their secondary calibration based on Givnish et al. (2015).

In the introduction and especially in the discussion, there is a lack of comparison with other studies dealing with diversification of East Asian forest-dependent lineages. Gamisch & Comes 2019 study on Bulbophyllum is an example of an article that could inform the author's discussion.

If any implications for Gastrochilus taxonomy can be derived from the new tree, these should be presented in the discussion, together with the acknowledgement of the limitations of the used taxonomic sampling.

Finally, the English of the manuscript must be improved. Some sentences, which I indicate in the enclosed revised version, were unclear to me and, therefore, I could not assess their scientific accuracy.

Author Response

Thanks very much for you constructive suggestions. The major revisions are summarized as follows. 1) we have additionally sampled four species of Angraecinae as outgroups. 2) We have used the split age of Angraecinae and Aeridinae (21.21 Ma) and the crown age of Aeridinae (16 Ma) to estimate divergence times of Gastrochilus

We have made a point-by-point response to your comments, which have  highly improved the quality of the manuscript.

Round 2

Reviewer 3 Report

The authors have made revisions to clarify many aspects of the study.  However, it is VERY concerning when the results shown for one version are changed from a previous version.  Either it means a serious mistake was made in one or the other version or that the data are not sufficently robust to support the conclusions.  The main point I refer to is in Fig. 2 in which the ancestral state for habitat was estimated (pie charts below the branches).  In the previous version, there are clear indication that the genus originated in tropical regions.  Now, it looks like there is higher probability of originating in subtropical regions.  It cannot be both ways.  This is very troubling.   Nonetheless, the fact that the pie charts in the revised manuscript still show considerable likelihood for tropical origin makes it clear that the authors should be more conservative with their interpretations of the results.  In other words, to claim that there is a southward migration of the genus is probably not warranted by the data (lines 298-300).  

The other concerning results shown are for the molecular clock analysis.  The fact that the strict molecular clock likelihood score is closer to 0 means that the log normal clock is NOT a better fit to the data.  Therefore, a strict clock should be used here.  But, there is something strange about the values shown.

Author Response

We sincerely thank you for your patience. We also thanks for your constructive comments, which improved the quality of the manuscript.

We have made a point-by-point response to the two comments in attached file. 

Reviewer 5 Report

I sincerely appreciate the authors’ work involved in this revised version. The calibration points are now clearly defined, and I am happy with the inclusion of some angraecoids in the outgroup. It seems the authors replicated the protocol of Givnish et al. (2015, 2016) by using one representative from each of the four angraecoid genera indicated in Givnish. However, the species of Angraecum chosen by the authors (i.e., Angraecum distichum) is not a “true” Angraecum, but it is rather best treated in Dolabrifolia (cf. Farminhão et al. 2021), falling in the same major clade as Aerangis and Diaphananthe. This fact resulted in the polytomy at the root of the tree, which is not desirable to set a calibration point. Should the authors had selected a true Angraecum (cf. Farminhão et al. 2021), which would then cluster with Aeranthes, this problem would have not emerged.

I am also happy with the fact that the authors added some references regarding the evolution of other orchid groups in the region.

The English is better now. Still, I have introduced some minor linguistic corrections in my new revised version.

Farminhão, J. N.M., Verlynde, S., Kaymak, E., Droissart, V., Simo-Droissart, M., Collobert, G., ... & Stévart, T. (2021). Rapid radiation of angraecoids (Orchidaceae, Angraecinae) in tropical Africa characterised by multiple karyotypic shifts under major environmental instability. Molecular Phylogenetics and Evolution159, 107105.

Author Response

We sincerely thank you for your suggestions.

In the revised manuscript, we replaced “Angraecum distichum by “Angraecum leonis” based on the reference (Farminhão, et al. 2021. Molecular Phylogenetics and Evolution,159, 107105.), and then we have updated all figures after re-performed all analyses on new data matrix. We also have revised all corrections on our ms. In general, these new results are consistent with our previous results.

Round 3

Reviewer 3 Report

I am not going to provide any more comments on this manuscript.  The authors do not appear to understand my comments about the molecular clock.  You and they could check the following paper for an example of what I mean regarding the likelihood scores for molecular clocks (see Table 1): https://academic.oup.com/mbe/article/32/11/2986/981260

The likelihood scores for the trees are negative numbers (although the authors show them as positive numbers in this latest version).  Therefore, the best fit models are SMALLER negative numbers.  In the case of this paper, that would be the strict clock.  There are other ways to evaluate model fit but the authors of this paper have chosen the LRT.

Author Response

Thanks.